# Benefits and Challenges of Making Data More Agile: A Review of Recent Key Approaches in Agriculture

**Elena Serfilippi [1,\*], Daniele Giovannucci [1] , David Ameyaw [2], Ankur Bansal [3], Thomas Asafua Nketsia Wobill [2], Roberta Blankson [2] and Rashi Mishra [3]**

1   The Committee on Sustainability Assessment (COSA), Philadelphia, PA 19147, USA
2   International Center for Evaluation and Development, Sakumono JWCP+XJ7, Ghana
3   GDi Partners (GDi), New Delhi 110065, India
\*   Correspondence: es@thecosa.org

**Abstract:** Having reliable and timely or ongoing field data from development projects or supply chains is a perennial challenge for decision makers. This is especially true for those operating in rural areas where traditional data gathering and analysis approaches are costly and difficult to operate while typically requiring so much time that their findings are useful mostly as learning after the fact. A series of innovations that we refer to as Agile Data are opening new frontiers of timeliness, cost, and accuracy. They are leveraging a range of technological advances to do so. This paper explores the differences between traditional and agile approaches and offers insights into costs and benefits by drawing on recent field research in agriculture conducted by diverse institutions such as the World Bank (WB), World Food Program (WFP), United States Agency for International Development (USAID), and the Committee on Sustainability Assessment (COSA). The evidence collected in this paper about agile approaches—including those relying on internet and mobile-based data collection—contributes to define a contemporary dimension of data and analytics that can contribute to more optimal decision-making. Providing a theoretical, applied, and empirical foundation for the collection and use of Agile Data can offer a means to improve the management of development initiatives and deliver new value, as participants or beneficiaries are better informed and can better respond to a fast-changing world.

**Keywords:** data innovation; data quality; household surveys

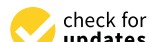

## 1. Introduction

There are many calls for better quality data and statistical modernization to guide sustainability investments and policies [1,2]. The broad ambition of the 2030 Agenda creates the need for an unprecedented range of reasonably high-quality statistics, at different levels: sectoral, subnational, national, regional and global [3]. Traditional data sources such as household surveys differ in terms of coverage, frequency, objective, timelines, and questionnaire design. This presents an important challenge for the monitoring of Sustainable Development Goals (SDGs) [4]. Further, in contrast to administrative data, household survey data have often been collected less frequently, and by multiple organizations and institutions, each with different focus and capacities. In some cases, the national surveys are implemented by governments, and in others, they are administered under the auspices of an international organization. Even this simple difference can generate substantial data gaps in terms of comparability, standardization, and the levels of disaggregation [4]. The high costs of traditional data gathering, associated with classical methods adopted in household surveys such as recall questions and self-reported measures, can frequently generate significant erroneous estimates of land [5], yields [6], farm labor [7,8], and fertilizer [9]. These facts signal the need for standardization and improvements in our monitoring systems and strategies to produce data in ways that are more reliable and more timely.

In this paper, we discuss recent data science developments that can transform not only the speed but also the accuracy and functionality of data collection and processing. Highlighting benefits and trade-offs of traditional non-agile data, including the time lags, high costs, and measurement complexities. The term non-agile is used simply to distinguish a contrasting approach; we do not imply a pejorative judgment. We present and characterize some emerging, technology-enabled solutions with Agile Data systems. We discuss the most recent tools introduced by the World Bank [9–14], USAID [15–17], CGIAR [18,19], World Food Program [20] and Acumen [21–23], focusing on how they present many of the *features* of agile approaches to data, and also how some do not manage to attain valuable agile *attributes*.

This analysis serves to characterize the main features of Agile Data, especially its ability to deliver real-time, high-quality data that can serve to improve interventions as they are unfolding. Costs are a salient feature and can be low because significant automation and standardization can be utilized across an array of context-appropriate technologies, (mobile, apps, chatbots, etc.) that reduce the need for enumerators travelling to the field. The approach is adaptive and with leading partners, such as the International Center for Evaluation and Development (ICED) in Africa and GDI Partners in South Asia, we are testing if it can accommodate a wide range of types of information related to program interventions, outcomes, compliance, well-being, and resilience. Similarly, within diverse applications such as those of the International Coffee Organization (ICO), the Gates Foundation's Sustain Africa (fertilizers), and the World Poultry Foundation (WPF), the processes are being adapted for easy and large-scale use.

Combined with new geospatial data technologies and artificial intelligence (AI), Agile Data offers place-specific insights into an array of concerns ranging from new practice adoption to human rights and deforestation. Agile Data also creates additional value when conducted in tandem with traditional technologies. With appropriate incentives, farmers tend to provide more timely and accurate responses during the data collection effort, generating more functional knowledge. This relational approach to data—what the Committee on Sustainability Assessment (COSA) with the Ford Foundation [24] calls Data Democracy—enables policy makers to better address the SDGs and pressing political agendas with timely and extensive monitoring frameworks to meet the demand for collection, processing, and dissemination of data by and within countries [4].

## 2. Materials and Methods

This paper builds on the most current work around agile, semi-agile and non-agile approaches to data to distill optimal practices and embedded knowledge already in use among data scientists and development practitioners. The vision of Agile Data was expressed by Owen Barder in 2013 [25] in reference to the software industry. This vision has been more recently adapted to the international development context [17,26]. In contrast with the waterfall model typically followed by the development community—meaning, based on implementation in sequence, just as waterfalls cascade in one direction—an agile, adaptive and interactive approach is based on feedback loops through performance metrics, as in Figure 1. By comparison with the waterfall model, the agile model is based on multiple rapid iterations of "design, build, test" to adapt the project design. In contrast, the waterfall model is often based around the implementation of a prepared master plan [17].

Combining the Agile project design and monitoring approach introduced by the literature with the innovation brought by digital technologies in data gathering, we obtain an Agile approach to data. In practice, what we define as Agile Data is a monitoring, evaluation and learning (MEL) data-driven approach that can improve outcomes and learning in development through an adaptive project design and monitoring combined with rapid deployment of surveys, data processing and analysis. In this approach, data are collected using short-duration or low-volume inquiries that can be conducted with high frequency and at relatively low cost.

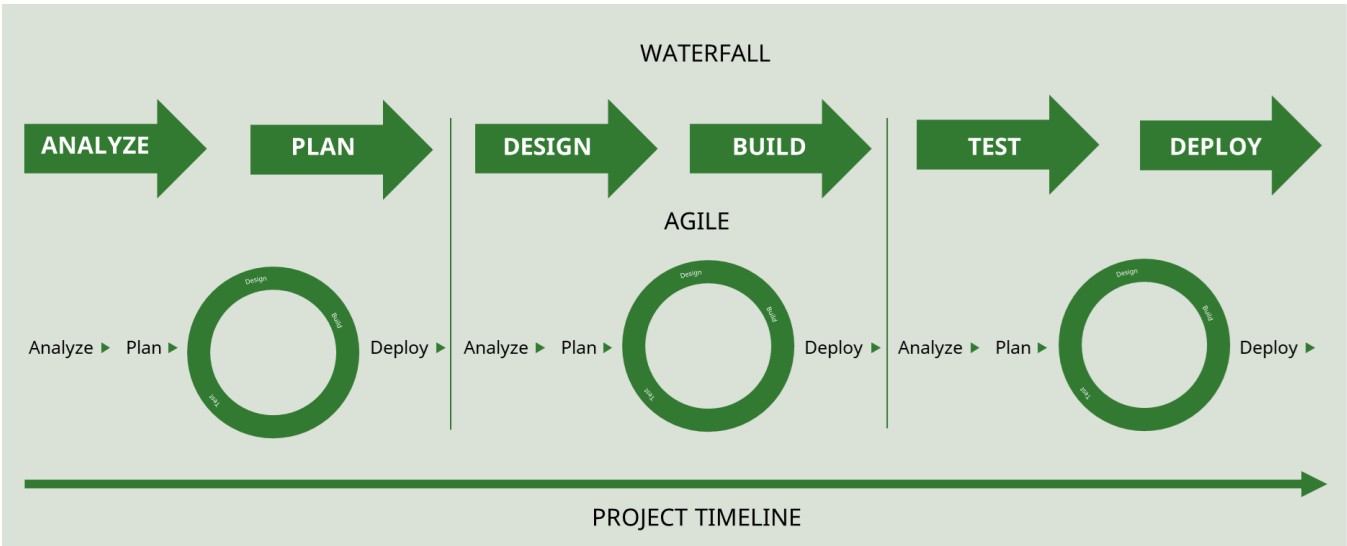

**Figure 1.** Waterfall versus Agile model. Source: USAID (2017) [17]. Reprinted/adapted from Ref. [17].

In Section 3.1, we compare the waterfall versus the Agile model, isolating characteristics of the Agile Data compared to the non-agile data through a systematic review of the literature. Section 3.2 of the paper contains a review of recent key lessons learned about Agile approaches to data in the field of agricultural development, highlighting the needs for statistical modernization and standardization. Through this review, the paper contributes to the characterization of an Agile Data system, helping development practitioners to build better data approaches to deliver greater knowledge efficiencies and potentially better impacts.

> **COSA Definition:** Agile Data is a Monitoring, Evaluation and Learning (MEL) approach that provides timely insights to facilitate adaptive learning and improve investment or intervention outcomes by rapidly deploying short-duration surveys that can be conducted at various fre-quencies and at relatively low cost. It applies targeted and context-appropriate field technologies such as IVR, apps, chatbots, or SMS and employs human or artificial intelligence to provide automated data validation, analysis, and feedback loops to users.
> Used in rural development programs or supply chains, it is configured to deliver higher-quality, real-time data, reducing survey fatigue among beneficiaries. It differs from most monitoring and evaluation by actively engaging data subjects more purposefully for more accurate information and mutual iterative learning during an engagement rather than after its completion.

## 3. Results

In the current complex and changing contexts of large-scale development programs and supply chains, we identify the potential benefit of shifting toward more Agile Data approaches and isolate six attributes that distinguish Agile from non-agile data utilizing a review of the recent literature. To support this hypothesis, we present evidence from field experiences of different organizations working on Agile approaches globally, highlighting the factors that are not yet developed and that will be necessary to take full advantage of these approaches. These include statistical modernization and standardization at both the semantic and structural data levels, together with the emergent opportunities to embrace the potentially considerable benefits of a more farmer-centric data approach. In particular, we illustrate the last point—an approach that we call Data Democracy—of how an Agile Data approach can build on a farmer's own first-hand information, and thus not only reduce the noise often present in multi-actor or multi-stage data channels (that gather, clean, and interpret), but also position farmers in a more direct learning and exchange function using a continuous flow of data throughputs to attain potentially unprecedented levels of understanding.

### 3.1. Non-Agile Data versus Agile Data

Much of the current data flow in development programs is traditionally structured as non-agile. It is a plan-driven approach with sequentially dependent steps such as analysis and planning, designing, pilot testing, and deployment for use [27]. The non-agile approach to data has been widely used and validated to collect accurate information on agriculture in complex environments including those characterized by small farm size, remote plots, multiple cropping systems, and poorly demarcated land boundaries [28,29]. According to Sourav et al. (2020) [30], large volumes of data are generated using traditional systems for data collection in agricultural contexts, but it is challenging to process the data using traditional data analysis. Non-agile approaches are disadvantaged by the costs associated with both the data gathering and the data analysis in the data generation process. Other examples in the literature show how the long process associated with data processing and analysis does not generate timely information, limiting the capacity of farm management methods to respond effectively to in-field variability in growing conditions or incorporate real-time information about weather conditions in managing agricultural activities [31,32].

The ultimate value in Agile Data for development is the Agile mechanism that enables teams to gather data quickly and more frequently. This approach allows us to understand key metrics faster, thus delivering quick learning through *feedback loops* and a greater possibility to respond to change and succeed [14,20,33–35]. An Agile approach allows for adapting questions through modules and across different data collection periods, iteratively building on learning from answers in early interviews. Recent studies in the field of extension services in agriculture, food security, and resilience show that an Agile data mobile phone-based approach to dissemination of information as a service for smallholders can have a positive impact in helping farmers to face shocks and stressors [36–39], improve nutrition checks and actions [40], promote farm management practices [41], deliver advice through an automated advisory service [42,43], and use speech-based services as a viable way for providing information to low-literacy farmers [44].

1. Measurement errors due to recall questions embedded in low-frequency surveys

At the micro level, non-agile data collection approaches could lack consistency and accuracy related to recall questions. Garlick et al. (2020) [45] argue that the recall method used for farm surveys overestimates farm labor per person per plot through recall bias that creates a countervailing effect on hours of farm labor at higher levels of aggregation, thereby overestimating the labor productivity of household-operated farms. Similar examples can be found in estimations of common measures of agricultural production and productivity [46]. The World Bank (2020) [10], using data from the Living Standard Measurement Study Integrated Surveys on Agriculture (LSMS-ISA) in Malawi and Tanzania, showed that with longer recall periods, farmers report higher quantities of harvest, labor, and fertilizer inputs, indicating the presence of non-random measurement error. According to Dillon et al. (2018) [47], Fraval et al. (2018) [7] and Kilic et al. (2021) [48] and Carletto et al. (2013) [49], self-reported measures of land size and yields generate bias in the estimation and create non-credible data observations. The unreliability of farmer-level observations, such as yield measurements, labor, and land, have decision-making implications for household land and thus substantially understate agricultural production and labor productivity [50].

Recent studies show that high-frequency phone surveys per single user are extremely useful for limiting the bias in the collection of agricultural data. An example is with regard to labor inputs or the harvesting of continuous crops such as cassava, for which the use of long recalls is highly inaccurate [51–53]; or for water quality measurement in the case of aquaculture management [54]; or for plot size and productivity [55–58]; or for enforcing labor contracts [59]. The recent integration of active artificial intelligence (AI) is further facilitating the interpretation of land use data and is making it more commonplace to access data that are no more than a few days old. Satellite data are also used to provide precise yield estimates. Benami et al. (2021) [52] indicate that remote sensing data and spatial modeling allow users to estimate crop yields as well as monitor them at scale at

the requisite frequency and timelines; hence, many researchers are exploring improved algorithms that enable measurement of agricultural yield from space [28,60]. Carter et al. (2017) [61] and Flatnes et al. (2018) [62] validated the use of maize and rice yield estimates from higher-resolution satellite data. A study conducted by McCarthy et al. (2021) [39] in Uganda compared maize yield estimates generated from satellite data and crop models against farmer self-reports (surveys), subplot crop cuts, and full-plot crop cuts. The results show that remotely sensed yields captured over half of the variability observed in the full crop cut data for pure stand (i.e., not intercropped) plots > 0.10 hectares. These results point to a promising possibility of eventually using inexpensive, publicly available earth observation data combined with crop models to characterize key field conditions such as yields and yield losses.

High-frequency panel remote surveys are also useful to track socioeconomic and health impacts of shocks such as COVID-19 and Ebola [50,63,64] and to enable the users to collect timely information on shocks, stressors, and associated resilience strategies [59,65,66]. Hoogeveen et al. (2020) [64] showed that when a systemic shock occurs, it tends to affect most of the different actors involved in the food supply chains (food producers, retailers, transporters, etc.) and prevents most of them from operating efficiently [64,67]. In the case of shocks, timely information would have contributed to a rapid communication between different actors of the supply chain and helped to diminish or prevent supply chain disruptions [5,19,50,68,69]. At the macro level where decision-makers focus on the bigger picture of policy implementation and impact, non-agile data approaches can weaken the policy response.

2. Low comparability at portfolio level due to limited interoperability (non-standardized metrics)

Non-agile data are usually focused on specific geographies and on assessing or reporting on an intervention or related issues of interest. Even though this approach enhances in-depth assessment within limited scopes and geographies, the definitions of measurement metrics for generating data are often inflexible for comparison with similar interventions and issues of interest such as other programs/projects, crop and livestock types, and geographical areas. The slowness to adopt common metrics has limited the ability to learn from field experience and has diminished the potential for more effective policymaking [70]. Blundo et al. (2018) [71] and Wanjala et al. (2017) [40] surmised that a conscious layout of an impact pathway, even ex ante, can facilitate identification of data needs for each anticipated milestone to inform prompt decision-making. However, using non-agile data approaches can be time consuming and cumbersome as tools to facilitate learning at key operational levels. Specifically, the integration of micro, meso, and macro levels becomes a particular challenge within the temporal confines of many programs. This fact adversely affects learning and use for decision-making along reporting hierarchies, as well as the ability to repurpose the data to meet other needs or outcomes [32].

According to Carletto 2021 [1] (p. 720), "the limited integration and interoperability of agricultural data has contributed to making today's agricultural data less relevant to tomorrow's policy challenges. Improving data integration and interoperability across data sources would greatly contribute to overcoming the limitations of individual data sources in achieving the temporal and spatial resolution needed for many applications". One area of data governance that is very important for developing e-agriculture (electronically facilitated information related to agriculture) is creating standards to harmonize the ways data are collected, processed, stored, and shared. To maximize the benefits of digital technology use, there needs to be some way to ensure the consistent collection, exchange, and dissemination of accurate information across boundaries, both sectoral and geographic. Without such consistency, there is a real risk of misinterpretation of information, and incompatibility of data structure and terminology [72].

3. Limited inclusivity due to lack of both a farmer-centric approach and open data principle

To ensure that a key development issue such as inclusivity (gender, youth, minorities, the poorest, etc.) is promoted as part of an intervention requires regular and consistent monitoring. Similarly, implementer and beneficiary satisfaction about inclusion needs to be tracked in real time. Saner et al. (2018) [67] argue for participation-based and inclusive monitoring to be fundamental components of managing the SDG implementation process to ensure transparency. Non-agile data approaches can be used to capture inclusivity, but often limit the extent to which decision-makers could be furnished with that data, so as to enhance timely responses to address those issues. According to Lamanna et al. (2019) [48] and UN (2020) [15], the common practice of interviewing the head of the household has generated significant bias and limited the progress toward achieving gender empowerment and inclusivity as per the SDGs.

According to COSA (2022) [24] and Schroeder et al. (2022) [43], Agile Data should also be farmer-centric or embrace Data Democracy principles, providing the right set of incentives to farmers to make them more engaged in the data collection process. In practice, there should be an open relationship and exchange of data with farmers in a mutually beneficial data ecosystem between the public and the private sector [39]. As suggested by Schroeder et al. (2022) [43], policy makers should employ a clear legal framework that recognizes "a general principle of access to privately held data of public interest". This claim of public interest is based on the public or collective contribution to the value of certain private data assets. These can include identifying and building on competitive spaces for sharing private sector data—where private actors realize the value of open data in promoting innovation, cost-sharing, and value chain efficiencies—and combining with other datasets for new or expanded insights. The public sector could also create public–private partnerships by, for example, co-financing research and development with private-sector firms.

4. High costs of conducting face-to-face surveys in the field

As with data collection in general, the issues related to the practical application of mechanisms to collect qualitative, quantitative, or mixed data are interrelated. Theobald and Diebold (2018) [66] posited that weaknesses in non-agile data collection and management include interface problems related to project planning; controlling, reporting and approval; contracting and budgeting; process requirements; tooling and infrastructure; coordination and integration; and staffing. In general, it is not easy to obtain good cost estimates on household surveys since funders tend to keep cost information confidential. Data for Development (2015) [73] compares the costs of different high-quality surveys based on computer-assisted in-person interviews (CAPIs). The analysis shows a per-survey cost that ranges from about USD 450,000 to 1,700,000, Table 1.

**Table 1.** Average cost per survey type (USD) *.

| Type | DHS | MICS | LSMS | Labor | Agricultural | Supp. |
|---|---|---|---|---|---|---|
| **Operations** | 800,186 | 716,040 | 1,235,852 | 331,204 | 1,117,303 | 319,002 |
| **Field** | 805,027 | 340,985 | 495,427 | 133,128 | 431,135 | 125,974 |
| **Total** | 1,605,213 | 1,057,025 | 1,731,279 | 464,333 | 1,548,438 | 444,977 |

"Operations" consist of training, transport, personnel and data processing. "Field" support refers to technical assistance, admin. and other costs. "Types" refers to DHS: Demographic and Health Survey; MICS: Multiple Indicators Cluster survey; LSMS: Living Standard Measurement Study; "Supp." Refers to supplemental surveys to measure progress toward SDGs. * Source: Data for Development: A Need Assessment for SDG Monitoring and Statistical Capacity Development.

Data for Development (2015) [73] and Dabalen et al. (2016) [33] highlight that "a typical complex, multi-topic household survey that is in the field for a year might cost around USD 140–150 per household—excluding technical assistance in sampling and data entry—and collect data on responses to roughly USD 3000 questions or about USD 0.06 per question, compared with USD 0.20 per question in a mobile phone survey".

Digital technology and new survey modes can reduce these costs. Agile Data relies on fast and easily accessible methods of data collection based on remote technologies,



such as computer-assisted telephone interviews (CATI), interactive voice response (IVR), mobile applications and sensors. Remote technologies seek to replace, at least partially, the face-to-face interviews with voice calls from live operators (CATI), SMS, or through pre-recorded messages (IVR) or mobile applications and chat applications to enable more frequent data collection for a lower cost. These technologies are more often combined with remote sensing collecting physical data to be integrated into the Geographical Information System (GIS).

Table 2 presents an analysis of attributes of the different technologies associated with the modes of survey administration. CATI and remote surveys in general are more cost-effective for data collection since operating mobile phone calls is usually cheap and can produce ready-to-analyze data in near real-time [14,33,74,75]. Utilizing both SMS and direct phone calls, preliminary results from a pilot initiative in Botswana suggest that phone assessments can provide valid information, under certain conditions, at a fraction of the cost of face-to-face interviews [76].

**Table 2.** Modes of administering survey and typical related attributes.

| Technology Mode | | | Key Attributes | | |
|---|---|---|---|---|---|
| Mode | Physical Set Up | Hardware Requirements | Time Saving—Low Costs | High Frequency | Feedback Loops |
| CAPI | Face to Face Operator | Mobile Phone/Computers | | | |
| CATI | Live Operator Call centers | Mobile phone | | | |
| SMS | Automated & Manual | Mobile phone | if manual | | |
| Mobile App | Automated | Smartphone or feature phone | | | |
| IVR | Pre-Recorded Messages | Mobile phone | | | |
| Sensors | Automated | Satellite | | | |

Note: Red = missing attribute; Green = attribute fully satisfied; Orange = attribute partially satisfied.

Data for Development (2015) [73] attempts to compare these costs with the cost of a standard face-to-face survey, but there are no obvious and easy ways to accomplish this given the differences in sample sizes, the frequency of data collection, the complexity differences of questions, and the number of questions per module. From a recent IPA (2020) study, it is clear that self-administered modes—IVR and automated SMS—cost less to implement than CATI because they do not require the same personnel costs (human interviewers and supervisors), as shown in Figure 2 elaborated by IPA (2020) [77]. Further advantages of IVR compared to other communication channels, such as SMS and most mobile phone applications, are that voice messages or surveys can be recorded in different local languages and accessed on demand, and farmers can easily follow the voice message even if they do not know how to read. Results from a study in Ghana conducted by CGIAR (2022) show that farmers are willing to use mobile phones to receive agricultural information [24]. However, they prefer voice channels over text, which may be related to the low education and literacy level.

The positive attributes of Agile Data approaches compared to non-agile data are summarized in Table 3.

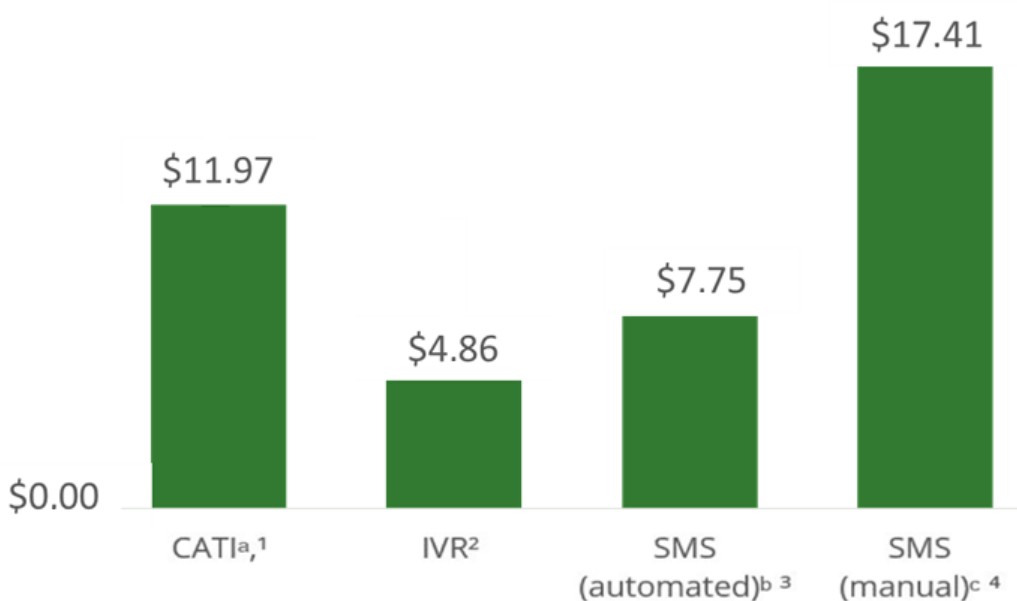

**Figure 2.** **Average cost per completed survey** (USD). **Source:** IPA (2020) [76]. Reprinted/adapted from Ref. [76]. [a] A portion of these estimates do not include fixed costs and underestimate total survey costs. [b] Automated SMS utilizes technology that can schedule messages rather than rely on human interviewers to send messages to participants [c] Manual SMS utilizes human interviewers to send messages to participants. [1] Kenya (32); Lebanon (43); Madagascar (7); Malawi (7, 49); Mozambique (53); Nepal (55); Peru (3); Senegal (7); Sierra Leone (42); South Africa (18); Tanzania (7); Togo (7). [2] Afghanistan (40); Bangladesh (19); Ethiopia (40); Ghana (24, 31, 54); Mozambique (40); Peru (3); Sierra Leone (42); Uganda (19); Zimbabwe (40). [3] Liberia (17); Peru (3). [4] Nepal (55).

**Table 3.** Typical positive attributes of Agile and Non-Agile Data in international development.

| | Agile | Non-Agile | Reference to the Literature |
|---|---|---|---|
| 1 | Diverse technology modes | Surveys administered face-to-face or by telephone = Higher costs | [3,4,12,33,73,77,78] |
| 2 | Short duration, and high frequency | Long duration of data gathering and processing = Less actionable knowledgeLow frequency = Measurement errors, non-timely information and greater attrition rates | [1,5,6,17,30,35,40,45,66,79–82] |
| 3 | Agile design and monitoring based on rapid feedback loops and adaptive behavior | Waterfall or linear management model is more static and less interactive, thus reducing flexibility and rapid learning or decision-making | [12,15,17,20] |
| 4 | Open data principles | Closed data ecosystem = Limited exchange of data to farmers and between the public and private sector | [74,83–86]; |
| 5 | Farmer-centric approach to Data Democracy | Limited ongoing farmer engagement | [20,24,43,48,87,88] |
| 6 | Interoperability | Limited integration between different data types and sources. Non-standardized metrics that challenge verification and can limit the topics and levels of analysis set in the beginning | [1,40,71,89] |

### 3.2. The Status of the Literature: Agile Data and Semi-Agile Approaches

In recent years, there has been a proliferation of semi-agile approaches to data. The term "semi-agile approach" stands for approaches with characteristics similar to those of the Agile approach to data, but with incomplete attributes. The World Bank, USAID, CGIAR, the World Food Program, Acumen, and many other development practitioners have introduced tools to inform decision-making in ways that were never possible before thanks to new digital technologies, ensuring sustainable development outcomes through greater efficiency, agility, and performance.

In the following section, we review some of these tools in relation to the survey modes and some of the six Agile Data attributes. The characteristics of each tool in relation to these agile attributes are summarized in Table 4.

**Table 4.** Semi-agile approaches to data collection in agriculture.

| Organization | Diverse Tech Modes | High Frequency | Short Duration | Agile Design and Monitoring | Farmer Centric | Open Data Principles |
|---|---|---|---|---|---|---|
| **World Bank** | CATI | RRPS: Multiple rounds LSMS-ISA: monthly | RRPS: 20 m LSMS-ISA: 20 m SWIFT: 7–10 m | IBM | | LSMS-ISA: RRPS: Interactive country dashboard |
| **USAID** | IVR, mobile app | RTD4AM: monthly or weekly | RTD4AM: rapid | RTD4AM MERL | RTD4AM | RTD4AM |
| **World Food Program** | SMS, CATI, IVR, chatbot, Facebook | mVAM | mVAM: rapid | mVAM | mVAM dashboard | mVAM |
| **CGIAR** | IVR | 5Q: Daily | 5Q: 15 min | | 5Q: Dashboard | |
| **Acumen** | CATI | | | | | |

Note: Red = missing attribute; Green = attribute fully satisfied; Orange = attribute partially satisfied.

First Attribute: Diverse technology modes. The technologies used to capture the data are different and may also include using advanced statistical methods to reduce required sample sizes and using tools that can support rapid data collection: cell phones and tablets for survey implementation, SMS, mobile apps and IVR technology for remote data collection, and geospatial imagery from satellites. For example, CGIAR's innovative 5Q approach, developed in 2015 and refined in 2021, uses smart-question trees (5Q-SQTs) to recall a farmer's perception, monitor the effects of implemented activities, or evaluate adoption, among others [18,75]. Questions are asked through IVR, and they are linked in a tree structure by branches and decision nodes, connecting a respondent-based answer choice to the subsequent question block. Another example is USAID's Real Time Data For Adaptive Management (RTD4AM) that has heavily invested in mobile platforms and IVR technology [40]. USAID's M-Posyandu project, for example, uses a smartphone application called M-Posyandu to use real-time mobile data to improve the efficiency and quality of nutrition service decision-making and achieve national nutrition goals [90]. USAID's Listening Post uses an interactive rural radio platform to provide broadcasts and radio mini-series on specific topics [91]. Listeners, mainly farmers, are then invited to participate in polls, ask questions, and offer opinions. The Listening Post project in Tanzania is a pilot initiative with funding from Bill and Melinda Gates Foundation and undertaken by Farm Radio International (FRI) [91]. In order to gather and analyze the mobile phone-based feedback, the project uses Uliza—a tool built with interactive voice response (IVR) by FRI. The system is built around IVR developed by Voto Mobile, and it enables listeners to vote in polls, leave messages, and request information.

Second Attribute: Short and high frequency surveys. The World Bank and Acumen mainly use CATI technology, but by introducing modifications to the survey length. These surveys allow for near real-time survey data collection and make it possible to quickly cover wide areas. For example, WB's Rapid Response Phone Surveys (RRPS) are quick

surveys administered through CATI to households, businesses or firms, with each interview typically lasting less than 20 min [9]. Further, WB's Survey of Well-Being via Instant and Frequent Tracking (SWIFT) is a low-cost, low-frequency survey (annual), to collect welfare information from project beneficiaries, as well as to monitor a project's contributions to extreme poverty and shared prosperity by providing timely feedback to project teams. SWIFT requires only 7–10 min for each household interview, a few minutes for processing the data, and costs less than USD 100,000 per country to implement [13]. In Tanzania, SWIFT is being used to fill a critical data gap in mobile penetration across income groups by combining questions on mobile phone uptake and usage with consumption estimates.

The World Bank recently transformed its well-known traditional multi-country and multi-round FTF Living Standards Measurement Study—Integrated Surveys on Agriculture (LSMS-ISA) program into a high-frequency monthly phone survey following the outbreak of the COVID-19 pandemic [10]. The proposed High-Frequency Phone Survey (HFPS) in each of the 5 LSMS-ISA countries (Uganda, Ethiopia, Nigeria, Malawi, Tanzania) will track the responses to and economic impacts of COVID-19 by conducting monthly phone interviews with a national sample of households that had been interviewed during the latest round of the LSMS-ISA-supported national longitudinal household survey and/or an alternative, recent, nationally representative, cross-sectional survey that may also be available. Each month, the HFPS households will receive a core set of questions primarily to capture the economic impacts of COVID-19, and these questions will be complemented by rotational questions on select topics that will be introduced each month and kept to an agreed length. Within the core set of questions administered in each country, a selection of these will be comparable across countries. The monthly interview with each HFPS household will not exceed 20 min.

Acumen, inspired in part by the principles emerging from the Lean Research Initiative that also included MIT D-Lab and the Fletcher School at Tufts University in 2015, developed an approach to what they termed LeanData [21,92,93]. This approach has been based on telephony to essentially reduce the time and costs of data gathering. For example, it captured information on poverty via the Progress out of Poverty Index (PPI—now Poverty Probability Index). They further developed Toolkits on Gender and Climate Resilience with applications for customers in a Resilient Agriculture Fund [22,23,94].

Third Attribute: Agile and adaptive design and monitoring. Most of the tools developed by the main development practitioners seem to have embraced the logic of the Agile project management over the waterfall model. Feedback loops are indeed present in the WFP's Mobile Vulnerability Analysis and Mapping Approach (mVAM), WB's Iterative Beneficiary Monitoring (IBM), USAID's Rapid Feedback MERL (RF-MERL), USAID's RTD4AM, and CGIAR'S 5Q approach. These approaches complement traditional ME methods by increasing the frequency of stakeholder consultation to understand how project activities are impacting, providing timely information for corrective action. USAID's RF-MERL in Tanzania, for example, has implemented regular "learning checks" in which all partners come together to reflect on the findings, brainstorm ways to refine implementation, and iterate accordingly, in order to strengthen community engagement in children's learning. USAID's M-Posyandu shows that through this mobile phone application, counselors can input monthly information about children and automatically process growth measurements. The system also flags nutritional risk, allowing counselors to tailor health messages for parents in real time. All measurements are stored in electronic health records that are available in real time or nearly real time at sub-district and district levels, where they trigger responses by health care officials and NGO staff. Counselors who used mobile phones were more likely to provide feedback on their sessions, and the system accelerated the process of nutrition data collection and improved data accuracy by 80%.

The Institute of Development Studies conducted a detailed research study exploring whether and how the USAID Listening Post could support adaptive management processes [95]. The research found that Listening Post has demonstrated its potential to collect real-time feedback from farmers that could be used to aid decision-making and improve

accountability in agricultural development initiatives, helping to ensure they are more responsive to farmers. WB's IBM has been tested for a wide range of topics, from school meals and fertilizer subsidies to free medical care, and in different countries, including Mali, Niger, and Nigeria. IBM's feedback mechanism led to notable improvements in project implementation: more students receive school meals, more farmers receive fertilizer vouchers, and more women have access to social protection than would have been the case without IBM. It helps to monitor project investments and the reach of beneficiaries, guaranteeing social inclusion. CGIAR's 5Q uses simple sets of questions, part of a logic-question-tree structure, implemented in multiple rounds, thus simplifying the burden for respondents while rapidly and with high frequency providing feedback to the project implementers.

Fourth and Fifth Attributes. Farmer-centric and open data principles. Although these tools often implement an agile feedback mechanism and deploy data in dashboards, such as WB's RRPS, CGIAR's 5Q and USAID's RD4DM, they show limits concerning open data principles, data ownership and a farmer-centric approach. The data infrastructure process, in the form of database ownership and data management systems, is often not clear, except for the case of WB'S LSMS, in which data are public. Further, experience from the field shows that it is difficult to share data back directly with respondents. The research conducted by IES found that USAID's Listening Post has demonstrated its potential to collect real-time feedback from farmers that could be used to aid decision-making. However, it also concludes that "closing the feedback loop"—ensuring that a farmer's comments, questions and concerns are responded to—is a challenge for the Listening Post. Some results have been reached by WFP through the Mobile Vulnerability Analysis and Mapping (mVAM). This tool is used to collect data on households' vulnerability and food security [20]. There are three modes of data collection embedded in this tool: CATI, through live calls from a call center, SMS surveys, and Interactive Voice Response (IVR). In Somalia, the operators who place outgoing calls also take incoming calls from beneficiaries all over the county. In the Democratic Republic of Congo, WFP set up an IVR system to respond to questions from beneficiaries. WFP is currently working on two fronts: webpage and chatbots. Through chatbots, respondents are contacted on Telegram via their smartphones and asked a series of questions about their food security and livelihood situations just as they would be by phone, SMS, or on other mVAM modalities. WFP is also working with Facebook's Free Basics platform. With Free Basics, people can access relevant information for free via their internet-enabled phones. WFP works in over 50 countries, and it is piloted right now in Malawi, where people can access weekly market price data, market news, and a polling function that allows users to take simple surveys and provide feedback that mVAM has collected on a website.

Sixth Attribute: Interoperability. There is so far limited clarity about how these tools ensure consistent data collection, exchange and the dissemination of accurate and standardized information. In some cases, when the data collection efforts are repeated over multiple years, as in the case of LSMS, the benefits of digital technologies are maximized since there are specific standards to harmonize the ways data are collected, processed, stored, and shared.

## 4. Discussion

As a primary goal of this article, we examined the changing pathways through which digital technologies and adaptive methods can accelerate the gathering and use of data, particularly for development objectives. Reviewing the recent applications of agile approaches in the field of agricultural development, we offer evidence to define the salient characteristics that distinguish Agile Data through six crucial attributes.

As developers and users of such approaches, we note that there are still limitations and challenges and, in this section, we discuss those and propose potential solutions that could be tested so as to further develop what we believe could be useful data tools, especially in difficult or remote regions.

### 4.1. Lack of Digital and Physical Infrastructure: Potential Bias and Solutions

First, it is clear that technological innovation is central to enabling such Agile Data approaches. Second, the low-cost and relative scalability allow access to much greater numbers of beneficiaries and thus permit a richer understanding and the ability to segment groups (by age, gender, income, etc.) to observe the effect of different treatments, capacities, or conditions. It can thus serve to assess the inclusivity of participation, particularly in regard to gender or minorities.

However, there is a substantial challenge manifest in different levels of access to digital technologies that, if not properly addressed, can exclude some of the most vulnerable and could thus manifest as a data or sampling bias and result in possibly deepening the digital divide [96–98].

For example, the potential of Agile Data to conduct high-frequency inquiries requires that data collection happen via a digital medium. However, the completeness or quality of data collected can be somewhat dependent on the accessibility of the respondents to mobile devices. According to Jarvis et al. (2015) [75], mobile-cellular subscriptions had grown in the five years prior to his study, especially in Asia and Africa. However, a gender gap still exists in low- to middle-income countries (LMICs) where women are on average eight percent less likely to own a mobile phone than men and use a smaller range of mobile phone services, such as SMS and internet access [99]. Mobile user data not surprisingly also show that the proportion of people who use smartphones to access the internet decreases with age and increases with educational attainment and household income [89]. In particular, literacy is crucial for the use of many digital technologies. Farmers in developing countries and smallholders in general may lack the skills and knowledge to reap the full benefits of digital applications that are available to them [100]. The fact that only a fraction of the population uses mobile phones, and this subpopulation may not have the same characteristics or behaviors as the population of interest, can easily generate a sampling bias [25,101,102]. In other words, the sample could systematically under-represent some groups, notably migrants, younger individuals, the poor, women, and people who do not have the skills or the capacity to use the technology offered [77,79].

Some studies have directly provided telephones to respondents in order to fill a critical data gap in mobile penetration across income groups and to understand how they access digital technology and use phones across socio-economic groups within the country [103,104]. Given the high costs of a single in-person survey, this can be cost-effective. The results have been used for evidence-based policy recommendations to increase access to mobile phone and internet technology for the poorest segments of the population.

Agile Data approaches can counter some of the exclusionary aspects of the digital divide and foster uptake and inclusivity with technologies that help reduce resistance with simpler intuitive interfaces [105]. Further, new technologies such as IVR and other voice-based applications could help to overcome some of the bias inherent in participants' levels of literacy. These types of efforts can help to overcome the lack of technological capacity among some users in the field and, therefore, facilitate adoption and use of these data approaches.

### 4.2. Adoption of Digital Technologies and Related Incentives: Toward a Farmer-Centric Approach Characterized by Data Democracy

A selection bias may occur based on the level of interest of the participants in the medium of data collection—especially with relatively novel mobile apps where the participants can tend to self-select. This opens some possibilities of unintended exclusions or even multiple inclusions of the same party, leading to sample frame errors and skewed results. The World Bank (2020) [11] proposes to use stratified sampling to overcome sampling bias and a system of incentives to increase participants' response rates. According to the World Bank (2020) [11], stratification based on forecast (ex ante) characteristics helps to balance the sample, and some of the literature notes that incentives for on-farm adoption of digital technologies are based on their perceived costs and benefits [43].

There is some consensus that monetary incentives, the most widely tested, increase response rates by reducing refusal rates, but do so with diminishing returns over time and even as the size of the incentives increases [75,82,106]. We note in a prior COSA and World Bank effort in Indonesia, that monetary incentives can be insufficient to ensure that accurate data are provided by respondents. Garbage in, garbage out is a clear concern that can overcome other efforts or methods to ensure data rigor. COSA and its partners, ICED and GDI, will test whether farmer-submitted data are more accurate when that same data are analyzed and offered back as useful benchmarks. For example, knowing one's input-use efficiency (e.g., labor or fertilizer costs per kilo or ton produced) relative to other farmers in the same zone and growing the same crops can offer valuable insights, but only if the data submitted to the algorithm are accurate.

Behavioral drivers of adoption, i.e., risk aversion and socio-economic characteristics such as land and farm size, together with the capacity of the technology to satisfy farmers' needs, are crucial features to be considered in the adoption of any particular digital technology [43,95,107].

Embracing a farmer-centric approach that directly benefits farmers and ensures their data ownership and access, offers potentially longer lasting opportunities to empower their active participation in the data effort. Farmers are often reluctant to adopt digital technologies due to lack of trust between them and third-party actors who collect, aggregate, and share data. In other words, unclearly defined data ownership, access, and control rights could lead to data misuse, eroding farmers' trust in digital technologies and discouraging their adoption [43]. The European Parliamentary Research Service and the German Agricultural Society both hold that for them, the farmer owns the data originating from his or her farm [47,74]. Although data ownership is important, we need to also guarantee crucial access to data and productive data use by farmers [108].

Perhaps the most valuable corollary feature of Agile Data is that it can incorporate a system of creative incentives to participate in the data collection efforts and to provide accurate data. Data providers or beneficiaries can be rapidly engaged with benchmarking and use of their data to tailor functional knowledge within a process known as Data Democracy. When functioning as open digital platforms and anonymized databases, they can help ensure that the voice of the farmer is heard directly.

*4.3. Data Protection Challenges: Need for Clearly Defined Data Rights*

Agile Data can contribute to more timely and more informed decision making with targeted management feedback loops that may include sensitive information.

However, since digital technologies can collect new types and large amounts of farm data, often with geotagging or other identifiers, it is important to limit access and safeguard farmer rights to their data. Laws addressing the ownership or use of data from digital agriculture are frequently either missing or inadequate, particularly in low and middle income countries. Farmers that use digital technologies may tend to share disproportionally more of their data, sometimes inadvertently, which can exacerbate privacy issues and reduce their control. Clearly defined data rights could encourage technology adoption, while unclearly defined data ownership, access, and control rights could lead to data misuse, eroding farmers' trust in digital technologies and thus potentially slowing or discouraging their adoption.

## 5. Conclusions

In this paper, we have offered a more systematic understanding and characterization of Agile Data based on an analysis of the evidence in recent literature and our own development of the concepts. This is part of an effort to help apply the principles more widely to improve how data can better serve farmers and rural communities as well as the organizations, such as supply chains, development organizations and governments, that rely on or invest in those farms and communities.

We have further specifically offered a definition of Agile Data and highlighted how an Agile Data approach can allow monitoring, evaluation and learning (MEL) to be based on consistent standardized data, undertaken regularly from the field for understanding individual projects, and to better target project beneficiaries through active feedback loops [31]. This approach works across categories, crops, and geographies, whether for food security, poverty reduction, climate adaptation, gender inclusion, resilience or income generation. It also offers timely and frequent access to low-cost information that can help overcome conditions of limited time and resources for data.

Nevertheless, we highlighted concerns related to the digital divide and potential failures to be inclusive that need to be actively considered if these approaches are to reduce those barriers to equity and advancement. We also caution that the proliferation of novel data approaches can be detrimental if not well-governed from the perspectives of both privacy and data protection. It is a topic that must be actively addressed by government, policy makers, and programs. Implementing appropriate data security measures will be important, as will inclusive communication to foster trust in the system that is necessary for its acceptance and growth. We propose that further research is needed in four key areas:

1. To understand the effect of these digital technologies on inclusivity
2. To determine data accuracy, provenance, and veracity
3. To what extent approaches can be designed as interoperable to accommodate advanced data approaches such as those employed by the LSMS and CGIAR centers
4. To better understand the potential and limitations of the range of benefits related to a farmer-centric approach including the progression to shared local or regional data eco-systems.

**Author Contributions:** Conceptualization, E.S., D.G., D.A. and A.B.; methodology, E.S. and R.B.; validation, E.S., D.G. and D.A.; formal analysis, E.S.; investigation, E.S.; resources, D.G.; writing—original draft preparation, E.S., D.G., R.B., T.A.N.W., D.A., A.B. and R.M.; writing—and editing, D.G. and E.S.; visualization, E.S.; supervision, E.S.; project administration, E.S.; funding acquisition, D.G. All authors have read and agreed to the published version of the manuscript.

**Funding:** This research was funded by Bill and Melinda Gates Foundation grant number INV-031160.

**Institutional Review Board Statement:** Not applicable.

**Informed Consent Statement:** Not applicable.

**Data Availability Statement:** Not applicable.

**Conflicts of Interest:** The authors declare no conflict of interest. The funders had no role in the design of the study; in the collection, analyses, or interpretation of data; in the writing of the manuscript, or in the decision to publish the results.

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
