# Peer review of "Benefits and Challenges of Making Data More Agile: A Review of Recent Key Approaches in Agriculture"

_sustainability, doi:10.3390/su142416480_

Round 1
Reviewer 1 Report
The article is interesting and valuable in terms of why and how Agile Data can be used for development. However, the paper does trend towards being VERY long and very descriptive. While descriptive may be appropriate given the topic, I want to encourage the authors to consider condensing some of the manuscript. Specifically, from 3.3.1 - 3.3.3 where the authors focus on use of Agile Data by various development organizations could be reduced. Is it necessary to organize this section by each group (e.g., USAID, World Bank), when the focus of the paper is on types of Agile Data? Could the authors consider reorganizing by types of Agile Data and then simply acknowledge which orgs are using that type? Otherwise, the publication is well-written and for anyone interested in the topic, very informative.
Author Response
Comment 1: Paper very long and very descriptive
Answer: Yes, we do agree. We reduced the paper to 19 pages and made it less descriptive. Many thanks for the valid suggestion.
Comment 2: Condensing specifically from 3.3.1 - 3.3.3 and re-organization of this section.
Answer: Many thanks for the valid suggestion. We re-organized the section by Agile Data attributes and we acknowledged which organization contributed to enhance that specific Agile attribute. Please see the attachment with the new version of the paper.

Reviewer 2 Report
There are many unfinished points throughout the manuscript. The entire article is about 38 pages of content, but nothing innovative is written in it. The manuscript has big lack of topic knowledge.
There are many editing errors where, for example, the drawings and the whole layout have not been properly prepared!!!
The discussion was not conducted in accordance with the guidelines of the journal. An expansion of literature, conclusions and discussions is required.
In this form, I advise against publishing the article in this journal.
English needs a lot of improvement.
Author Response
Comment 1: There are many unfinished points throughout the manuscript. The entire article is about 38 pages of content, but nothing innovative is written in it.
Answer Comment 1: We have revised the full article to address continuity, length and consistency. We have reduced it by about half with 19 pages of text. As to "innovation", this is offered as a reasonably comprehensive review of Agile approaches emerging in recent years. If, however you have seen very similar content such that this would be considered duplicative of other good work out there, then by all means please do note any author or journal as we would love to have access to any such study to help inform and improve our work.
Comment 2: The manuscript has big lack of topic knowledge.
Answer Comment 2. It is difficult to address due to the lack of specificity in this comment. Although it is not necessarily indicative of topical knowledge, perhaps we can offer a proxy for some "topic knowledge". Collectively, the team has experience as the Director of M&E for the largest agriculture program in Africa, a decade at the World Bank, teaching at University level, and well over a hundred publications including in top tier journals, and yes some PhD's from notable US and European Universities. So, we would be happy to discuss or revert on the topic knowledge you would like to have addressed, if you would be so courteous as to inform us of that.
Comment 3: There are many editing errors where, for example, the drawings and the whole layout have not been properly prepared!!!
Answer Comment 3: We apologize for the result of a collective effort not perfected and we have now applied the skills of a professional copy-editor and re-formatted. We do appreciate the call to improve.
Comment 4: The discussion was not conducted in accordance with the guidelines of the journal. An expansion of literature, conclusions and discussions is required.
Answer Comment 4: We have expanded the literature, and changed the section with discussion and conclusion accordingly. We hope it adheres to the guidelines of the journal. We understood that the conclusion section is not compulsory in the Journal and the discussion section could be merged with the result section. We used the discussion section to show the main challenges/limitations of the work around Agile Data, while the result section in now more clearly and more tersely discussing the results and how they can be interpreted given the contextual perspective of previous studies and, of course, the working hypotheses for the paper. From the guidelines we understood:
- "The findings and their implications should be discussed in the broadest context possible and limitations of the work highlighted. Future research directions may also be mentioned. This section may be combined with Results."
Comment 5: In this form, I advise against publishing the article in this journal.
Answer Comment 5: We hope you may reconsider, given the substantial changes.
Comment 6: English needs a lot of improvement.
Answer Comment 6: Yes, agreed. We thought it was imperfect but understandable for most whose first language is not English - apparently we were wrong and have had a native speaker do the editing work. Sorry for the inconvenience or sub-par communication this has caused.
Reviewer 3 Report
Point 1: Title of the manuscript needs rework. It does not reflect that the manuscript is a review article. Additionally, “State of ...” does not convey whether the article would discuss ‘state-of-the-art’ techniques in terms of agriculture/farming (since the special issue is focused on farming 4.0) or ‘state’ of agile/non-agile approaches in general – the in-text content suggests the later. Finally, “approaches to data…” lacks clarity for the reader – are we talking about data generation, assimilation, dissemination, analysis or knowledge generation.
Point 2: Better partitioning is a must - sections, subsections, topics and sub-topics should have a logical connectivity – which seems missing in a lot of places. ‘Results’ section needs to be reorganized to maintain a logical flow of information. Ex: details pertaining to models/methods can be moved to a different section as case studies, while results should focus on methodological differences between agile and non-agile frameworks.
Point 3: Writing seems inconsistent – There is clear contrast between writings within the article. Some sections/topics are well written (you won’t see any int-text markings or comments from my side), while some are too complex and seems mixed up. Please go through the article and fix such variability and typos (at a few places).
Point 4: Please add tables to compare agile vs non-agile methodology. Please also add a table that summarizes the studies/articles reviewed in this work. Add different attributes such as survey duration, frequency, cost estimates, technology and references (or any relevant parameters) – such that the manuscript is more readable and easier to follow. Additionally, please change Figure 3 to a more informative image – connectivity or interdependence between different attributes of agile that follow the iterative behavior.
Point 5: Figures from Appendix, such as B1 should be moved to methodology or introduction section. By doing this, we can focus on novice readers and can formally introduce the notion of agile data.
Look-out: Please have a logical connectivity between sections and sub-topics. The article is difficult to follow through.

Author Response
Answer to Point 1: We changed the title of the paper making explicit that the paper reviews studies related to the field of Agile Data applied to agriculture. This is in line with the topic of the special issue. Concerning the term “approach” we use this term to describe the entire data gathering and processing combined with agile design and monitoring. We clarify this concept in the methods section.
Answer Point 2: We revised the result section in accordance with your suggestions. Many thanks for the precious advice.
Answer Point 3: We apologize for these inconsistencies. We reviewed the writing and further edited the article ensuring consistency.
Answer Point 4: We would like to thank you for the excellent suggestion. We added the Table comparing Agile and Non-Agile approaches to data with a reference to the main literature. This Table is now replacing old Figure 3.
Answer Point 5: We have followed this suggestion. Many thanks.
In general, we have fully revised English and formatting, improving logical connectivity between sections.
Answers to other comments can be found in the paper: Please see the attachment

Reviewer 4 Report
The authors of the paper ”Making Data More Agile and Non-Agile: A Review of Recent Key Approaches to Datain Agriculture” present a topic relevant especially for rural areas, highlighting technological advances, as well as the "differences between traditional and agile", including research in agriculture carried out by institutions international organizations involved, especially since this issue is a priority in many rural areas at European and global level.
The concepts, bibliographic sources and citations are appropriate within the work, the authors trying to capture specialized publications "the Agile Data vision was expressed by Owen Barder in 2013 [11] with reference to the software industry. This view has been adapted more recently to the context of international development [99].”
The research methodology is simplistically presented, the authors highlight the Argile model in the research process, being considered by the authors as an effective method of "Agile approach to project design and monitoring with the innovation brought by digital technologies in data collection..combined with the rapid implementation of surveys, data processing and analysis".
The results of the paper are oriented towards the application direction, highlighting the contribution "to defining the contemporary dimension of data and analysis which can contribute to more optimal decision-making" and the results obtained confirm the thesis regarding the process of "improving management . .it can respond better to the diverse needs of a rapidly changing world". However, we suggest the authors of the work to highlight their personal scientific contributions, the results of the study, to the specialized scientific literature.
The conclusions presented by the authors of the paper capture the basic elements of the paper's thesis, namely "the development and adoption of digital solutions in agriculture". However, I suggest the authors of the paper to present the limitations of the study in a scientific and not just an applied sense, as well as the implications of the research team in the continuation of research in the specialized scientific field. Moreover, as I presented for the research results, highlighting the scientific contributions to the specialized scientific literature.
We congratulate the research team for the work done, and after reviewing the aspects mentioned above, especially the results and conclusions chapters, we propose the work for acceptance.
Author Response
Dear Reviewer,
Many thanks for the precious feedback. We have changed Results, Discussion and Conclusion sections accordingly to your suggestions.
We attach a copy of the Manuscript with Track Change On to easily recognize the modifications made. Please kindly determine if our response is adequate to the modifications suggested.
Many thanks again for your support.
Warm Regards
Elena Serfilippi

Round 2
Reviewer 2 Report
Unfortunately, the article has not been corrected following the reviewer's comments. There are many unfinished points throughout the manuscript.
Author Response
Dear Reviewer,
Unfortunately we did not receive any precise feedback neither this time or the other time. The comments have been quite vague to be addressed, except the ones concerning editing and formatting. We have revised again editing and formatting. Please see the attachment for the revised version of the paper.
Best

Reviewer 3 Report
Check the updated manuscript for new comments and suggestions: 1. Please check and maintain text indentation. 2. Typos and suggested change in language at a couple of places. 3. Please update subsection 3.2 (inline with Table 4) as suggested in the update manuscript.
Author Response
Dear Reviewer,
Many thanks again for the precious feedback. We have revised the paper accordingly to your suggestions. Please see the attachment.
It has been a pleasure to review the paper with your inputs. Many thanks again.
Warm Regards
Elena Serfilippi
